# A flexible formula for incorporating distributive concerns into cost-effectiveness analyses: Priority weights

Øystein Ariansen Haaland[1]*, Frode Lindemark[1], Kjell Arne Johansson[1,2]

**1** Bergen Centre for Ethics and Priority Setting (BCEPS), Department of Global Public Health and Primary Care, University of Bergen, Bergen, Norway, **2** Department of Addiction Medicine, Haukeland University Hospital, Bergen, Norway

* oystein.haaland@igs.uib.no

## Abstract

### Background

Cost effectiveness analyses (CEAs) are widely used to evaluate the opportunity cost of health care investments. However, few functions that take equity concerns into account are available for such CEA methods, and these concerns are therefore at risk of being disregarded. Among the functions that have been developed, most focus on the distribution of health gains, as opposed to the distribution of lifetime health. This is despite the fact that there are good reasons to give higher priority to individuals and groups with a low quality adjusted life expectancy from birth (QALE). Also, an even distribution of health gains may imply an uneven distribution of lifetime health.

### Methods

We develop a systematic and explicit approach that allows for the inclusion of lifetime health concerns in CEAs, by creating a new priority weight function, $PW = \alpha + (t-\gamma) \cdot C \cdot e^{-\beta \cdot (t-\gamma)}$, where t is the health measure. PW has several desirable properties. First, it is continuous and smooth, ensuring that people with similar health characteristics are treated alike. For example, those who achieve 50 QALE should be treated similarly to those who achieve 49.9 QALE. Second, it is flexible regarding shape and outcome measure (i.e., caters to other measures than QALE), so that a broad range of values may be modelled. Third, the coefficients have distinct roles. This allows for the easy manipulation of the PW's shape. In order to demonstrate how PW may be applied, we use data from a previous study and estimated the coefficients of PW based on two approaches.

### Conclusions

Equity concerns are important when conducting CEAs, which means that suitable PWs should be developed. We do not intend to determine which PW is the most appropriate, but to illustrate how a flexible general PW can be estimated based on empirical data.

**Data Availability Statement:** All necessary data and code to reproduce are available at https://github.com/oeh041/Priority_weights and https://doi.org/10.5281/zenodo.3256151.

**Funding:** This work was funded by the Bill & Melinda Gates Foundation through the Disease Control Priorities Ethiopia (DCP-Ethiopia) project grant to the University of Bergen and Harvard T.H. Chan School of Public Health (grant number: OPP1162384). The funder had no role in study design, data collection and analysis, decision to publish, or preparation of the manuscript.

**Competing interests:** The authors have declared that no competing interests exist.

# Introduction

Prioritisation of limited health care resources involves saying no and yes, hard ethical problems and reasonable disagreements. In health economics, cost-effectiveness analysis (CEA) is used to identify the most efficient allocation of health care resources. Within such a framework, informed decisions based on explicit CEA rankings can be made. However, CEAs have been extensively criticised for not being sensitive to fair distribution of health benefits, and there are other important equity concerns which may be considered, and few standardised methods are available to directly adjust incremental cost-effectiveness ratios (ICERs) [1–3]. CEAs are used to inform policy decisions on the introduction or reimbursement of new technologies or pharmaceuticals in countries such as the UK, Norway, Sweden, the Netherlands, New Zealand and Australia, in some cases including considerations beyond CEA through concepts like severity or burden of disease [4–14].

Typical health metrics in CEAs are quality-adjusted life years (QALYs) gained and disability-adjusted life years (DALYs) averted [15]. These aggregate measures take into consideration not only the life expectancy (LE) of a group, but also the quality of life. Interventions and health programs are typically ranked by ICERs, where those that maximize QALYs gained or DALYs averted are favoured. However, such rank-orders of interventions based on ICERs have been criticized because they disregard concerns for those who are worst off [16]. There are good reasons to give higher priority to individuals and groups with impaired health [16]. As opposed to health maximisation, one could aim for equal distribution of health across all individuals in society. This would require that resources should always be directed at people with the most impaired health, so as to minimize inequality in health outcomes, even if this means that the total health in the population will be reduced. People's intuition about what is fair may lie somewhere between strict health maximisation and strict equality. A compromise between these perspectives is that health maximization is an important goal, but health gains to people with impaired health are assigned more weight than similar gains to people with better initial health [17]. This "in between" position is called prioritarianism and has been defended by the philosophers like Derek Parfit [18] and Matthew Adler [19]. According to prioritarianism, a cheap and effective intervention directed at the healthiest may be preferred to an expensive and ineffective intervention directed at those who have worse health if left untreated. Still, if cost and effect were equal, the allocation of resources should be directed towards people with more impaired health if left untreated. Finally, when comparing competing health programs, studies have found that people favor priority to worse off, and they favor interventions that benefit those that are worse off over slightly more cost-effective interventions that benefit better off groups [20, 21]. Such equity concerns are too important to leave out from standard CEAs.

One may also argue that non-health concerns, like age, indirect benefits, productivity, and financial risk protection, are also important regarding equity. However, this paper focuses on health related concerns. At the time of a prospective health intervention, concerns for health distribution can be roughly divided into two perspectives: The first perspective is forward looking and is only concerned with the distribution of the health gains, typically measured in QALYs or DALYs. E.g., one may think that one QALY gained for 50 people is better, equal or worse than 50 QALYs gained for one individual. The second perspective is dealing with the distribution of health over a lifetime, often measured as quality adjusted life expectancy from birth (QALE) [22, 23]. E.g., consider two individuals who may benefit from some treatment. One has QALE = 80 without treatment and 81 with, and the other has QALE = 20 without treatment and 21 with. Their gains would be equal, but the second is expected to get much less lifetime health. Hence, if an equal distribution of lifetime health in the population is a priority,

one would value the set {80, 21} above {81, 20}. The view that expected lifetime health is what matters draws support from the literature on health priorities and some empirical studies [24–28]. Note that although QALE often increases with age, a high age does not necessarily correspond to a high QALE. E.g., the QALE of an old person who has experienced decades of chronic illness may be far less than that of a healthy child. In other words, current age is not a good measure of QALE, and weights based on age are therefore different than weights based on lifetime health. In this paper we focus on applications of the lifetime health perspective.

Our aim is the development of a systematic and explicit approach to empirically include these equity concerns in the framework of CEAs, by creating a new priority weight function. The function should be smooth and continuous in order to treat people with similar characteristics alike. Further, it should be flexible regarding shape and outcome measure (e.g., cater to other measures than QALE) to be able to encompass a broad range of values. Finally, coefficients should have distinct roles, to ensure that different functions can be easily compared, and allow for easy manipulation of the function's shape.

## Methods

### Moving from utility curves to priority weights

A social welfare function (SWF) is a function that represents the sacrifices that society is willing to make to promote a more equitable health distribution. The input should be a health distribution (individual QALYs, DALYs, etc.), and the output should be a real number, so that different distributions can be easily ranked. An SWF incorporates trade-offs between total health gain and health inequality in a population. Well-known SWFs are the Gini index and the Atkinson index [29, 30]. Both include a parameter that indicates the degree of inequality aversion, ranging from 0 to infinite. Such SWFs are data hungry, because complete data about the health or health gains for an entire population are needed. Often the data required do not even exist. A different set of SWFs is the one based on the aggregated QALY model. These utilitarian SWFs have the form

$$W(t_1, \ldots, t_n) = \sum_{i=1}^{n} u(t_i),\tag{1}$$

where $t_i$ is the number of QALYs received by individual i (i = 1,...,n), and the utility function $u(t)$, defined over $t_i$, is positive and monotonously increasing ($u(t) \geq 0$, $du/dt > 0$). If $u(t)$ is concave ($d^2u/dt^2 < 0$), W represents societal preferences for equality. Equality indifference yields $u(t) = t$, and reduces W to be the sum of the QALYs received across all individuals in society. Finally, if $u(t)$ is convex ($d^2u/dt^2 > 0$), society is assumed to prefer inequality. The function $u(t)$ has been the focus of several papers [29, 31–34]. As opposed to the Gini and Atkinson indexes, calculating $u(t)$ does not require complete information about the health or health gains of an entire population $(t_1,...,t_n)$. We refer to Rodriguez and Pinto for a more detailed discussion on $u(t)$ [34].

However, $u(t)$ does not account for the distribution of lifetime health. We approach this problem by considering priority weight functions (PWs) instead. These are functions that adjust the weight of health gains according to health related equity concerns, such as health achievement in a lifetime perspective. A general form of an SFW based on PWs is

$$W(s_1, \ldots, s_n, t_1, \ldots, t_n) = \sum_{i=1}^{n} F(s_i, t_i),$$

where $t_i$ is the health level of individual i at the time of intervention, $s_i$ is the health gain, and

$$F(s_i, t_i) = \left| \int_{t_i}^{t_i + s_i} PW(x)dx \right|.\tag{2}$$

The vertical bars denote that the absolute value is to be considered, and x is a dummy variable used for integration. As an example of how (2) may be calculated, consider a sick individual with a health level without intervention of QALE = 60, and an intervention with a gain of 3 QALYs. The priority weighted gain of the intervention is now $F(3, 60) = |\int_{60}^{63} PW(x)dx|$.

Although the literature is lacking, one candidate for PW is the age-weight function of the Global burden of disease (GBD),

$$PW_{GBD}(t_i) = C \cdot t_i e^{-\beta t_i}. \tag{3}$$

Here $t_i$ is the age of individual i [35]. As mentioned, age weights are beyond the scope of this paper. Still, by letting $t_i$ denote the health gain or lifetime health of individual i, (3) becomes a PW with quite an appealing form, although lacking in flexibility. A lack of flexibility would also be a problem if one would use a linear PW, such as

$$PW_{linear}(t_i) = a + b \cdot t_i. \tag{4}$$

Flexibility is an important property of a PW, as it should be able to reflect a wide range of health related equity concerns. Further, if the coefficients of the PW have distinct roles, they can easily be adjusted to accommodate such concerns. E.g., one may wish to set an upper limit for the maximum weight, or set the maximum weight at a fixed value of t. Also, if the PW is fit to empirical data on people's preferences, the roles of the coefficients will help to disentangle different concerns of society. One may, e.g., wish to estimate both the t for which the PW reaches its maximum, and what the maximum is. This is not straightforward in (3). While these concerns could be addressed using (4), a linear PW falls short in other aspects. E.g., a hump is useful if one wants to give lower weights to health gains benefitting those with a very low QALE (typically children born with severe conditions). If one wishes to do the opposite, and prioritize health gains to those who are expected to achieve very little lifetime health, PW may peak at t = 0, and then drop at a faster rate when t is close to 0 than when t is higher. A PW should also be twice differentiable with respect to t, so that it is continuous (no jumps) and smooth (no breaking points), ensuring the approximately equal treatment of individuals with similar values of t.

## A priority weight function

In this section we present a new flexible PW with coefficients that have distinct roles, which is suitable in a lifetime health framework via (2). The formula is as follows:

$$PW_1 = \alpha + (t - \gamma) \cdot C \cdot e^{-\beta \cdot (t - \gamma)}. \tag{5}$$

Setting $\alpha = \gamma = 0$ reduces (5) to (3). Setting C = 0.1658 and β = 0.04 we get the weights shown in Fig 1A. Panels B and C in Fig 1 show how $PW_1$ can be altered by manipulating C and β. As seen, C has the effect of inflating or deflating $PW_1$ (Fig 1B). For β>0, it can be shown that (5) ultimately approaches $\alpha$ with increasing t. We see that for larger β's this happens more quickly than for smaller β's (Fig 1C). When β≤0, (5) increases with t, and never approaches $\alpha$ at all.

The intercept α of (5) shifts the curve up or down (Fig 2A), whereas γ provides a right-left shift (Fig 2B). As is shown in Fig 2B, it is possible to ensure that the maximum is at t = 0, making (5) strictly decreasing. Note that C is interpreted slightly differently for (5) than for (3). For (5), changing C will affect the relative weights between different values of t unless α = 0. Also, C≤0 is not a problem for (5), because α can be adjusted to make sure that $PW_1$>0 for all

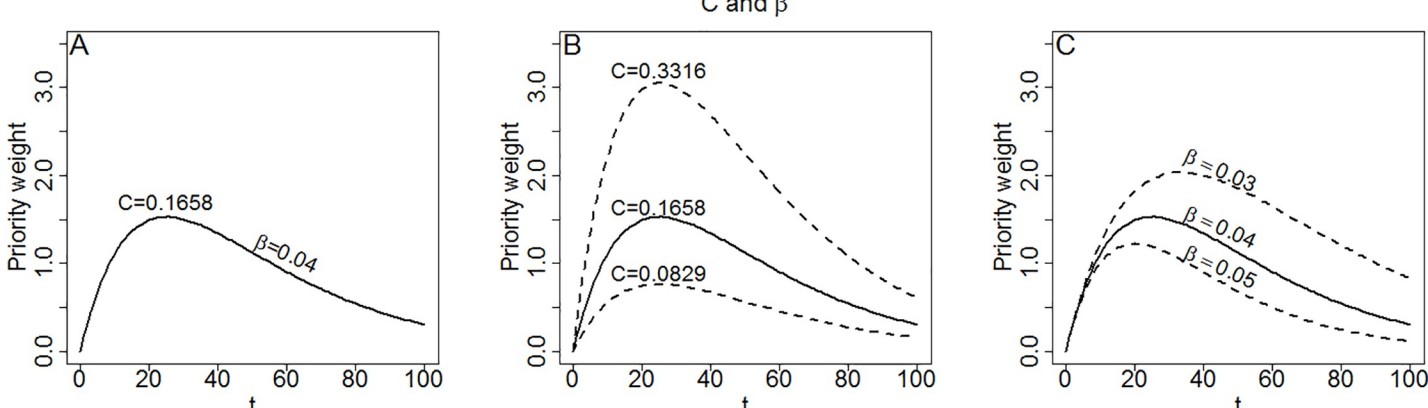

**Fig 1.** Properties of the coefficients of (5) when $\alpha = \gamma = 0$. Panel A: Priority weights according to t. Panel B: Altering C causes an inflation or deflation. Panel C: Altering $\beta$ changes the "pointyness" of the hump, causing the peak to become more or less pronounced.

relevant t's. A negative C simply causes (5) to be reflected through the line $PW_1 = \alpha$, or in other words, turned up-side-down. If C = 0, (5) reduces to the line $PW_1 = \alpha$, corresponding to inequality indifference. This is also true if $\beta$ approaches infinity.

As demonstrated in Figs 1 and 2, manipulating the different coefficients of (5) allows us to flexibly change its shape. E.g., if we want to ensure that people with a certain value of t are given a certain weight (anchoring the curve), we may simply change $\alpha$. Letting

$$\alpha = \alpha_0 - (A - \gamma) \cdot C \cdot e^{-\beta \cdot (A - \gamma)}, \tag{6}$$

anchors the curve to $\alpha_0$ at t = A. Further, if we want to set a maximum weight, so that max

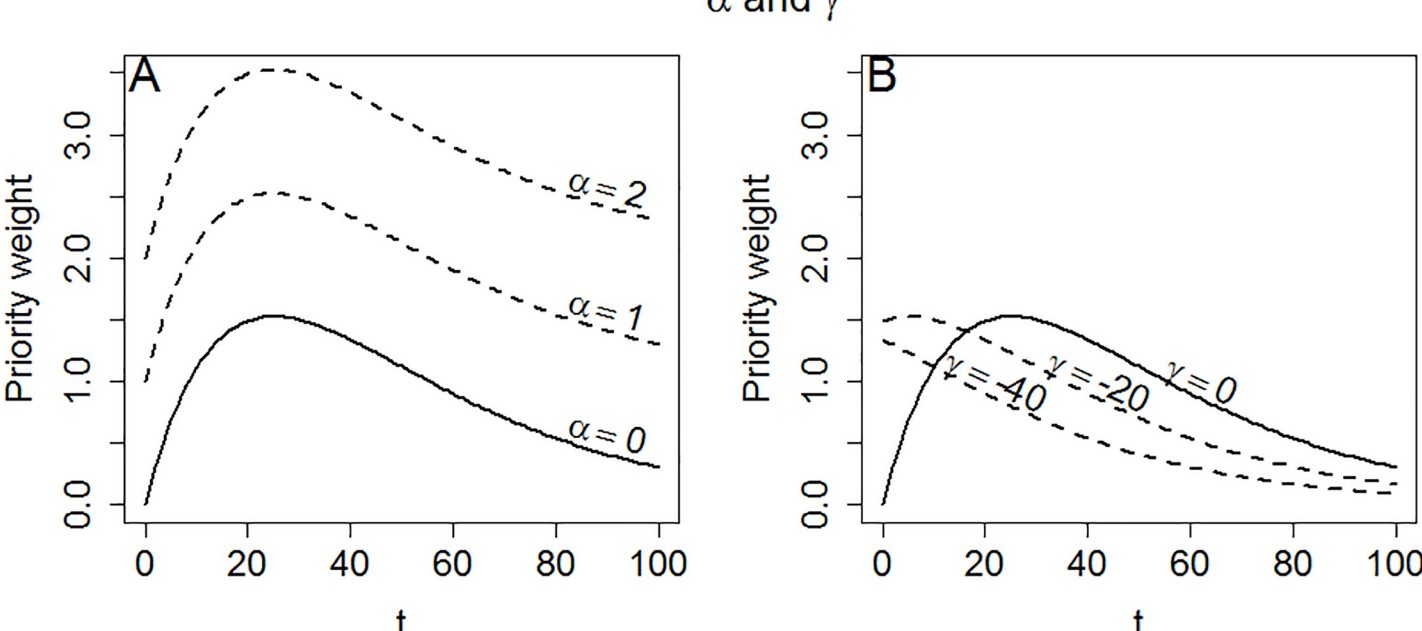

**Fig 2. Properties of the coefficients of (5) when C = 0.1658 and $\beta$ = 0.04.** Panel A: Altering $\alpha$ causes an upward or downward shift. Panel B: Altering $\gamma$ causes a left or right shift. Note that one would typically chose a PW where the maximum weight was at zero.

$(PW_1) = w_m$, we let

$$C = \beta \cdot (\alpha - w_m) \cdot \frac{e^{\beta \cdot (A - \gamma) - 1}}{\beta \cdot (A - \gamma) - e^{\beta \cdot (A - \gamma) - 1}}. \qquad (7)$$

If $\alpha_0$, A, $\gamma$, $w_m$ and $\beta$ are given, we can treat (6) and (7) as two linear equations with two unknowns, and solve for C and $\alpha$. Fig 3A shows (5) anchored to 1 at t = 80 for different values of $w_m$.

As mentioned, manipulating $\gamma$ will shift (5) right or left. It may be of particular interest to identify the shift needed to ensure that the maximum is moved to t = 0. This means that the slope is flat at t = 0, and gradually becomes steeper, so that gains for most individuals with poor lifetime health are given weights close to $w_m$. Differentiating (5) with respect to t, we find that the maximum is when t = 1/$\beta$. Hence, letting $\gamma = -1/\beta$ ensures that the maximum is at t = 0 (Fig 3B). Differentiating (5) once more with respect to t, the t for which the slope is the steepest can be identified as 2/$\beta$. Therefore, letting $\gamma = -2/\beta$ causes (5) to be at its steepest when t = 0. In such a scenario, only gains to people with very poor health are given weights close to $w_m$ (Fig 3B).

Note that although varying $\alpha$ without changing the other coefficients just shifts $PW_1$ up or down without changing its shape (Fig 2A), fixing the other coefficients in this manner is not something one would typically do. In order to keep predetermined properties of $PW_1$, such as anchoring, a change in $\alpha$ would prompt a change in one or more of the other coefficients.

So far the mathematical properties of the coefficients in (5) have been presented, but the policy implications and conceptual meaning of these properties may not be obvious. The next paragraphs are dedicated to the discussion of these implications. First, we consider $\gamma$. If $\gamma > -1/\beta$, it is possible that $PW(t_1) < PW(t_2)$ for $t_1 < t_2$. This makes sense if one thinks that small gains to individuals with almost no lifetime health should be worth less than a similar gain to an individual with more lifetime health. As mentioned, this could pertain to treatment of children

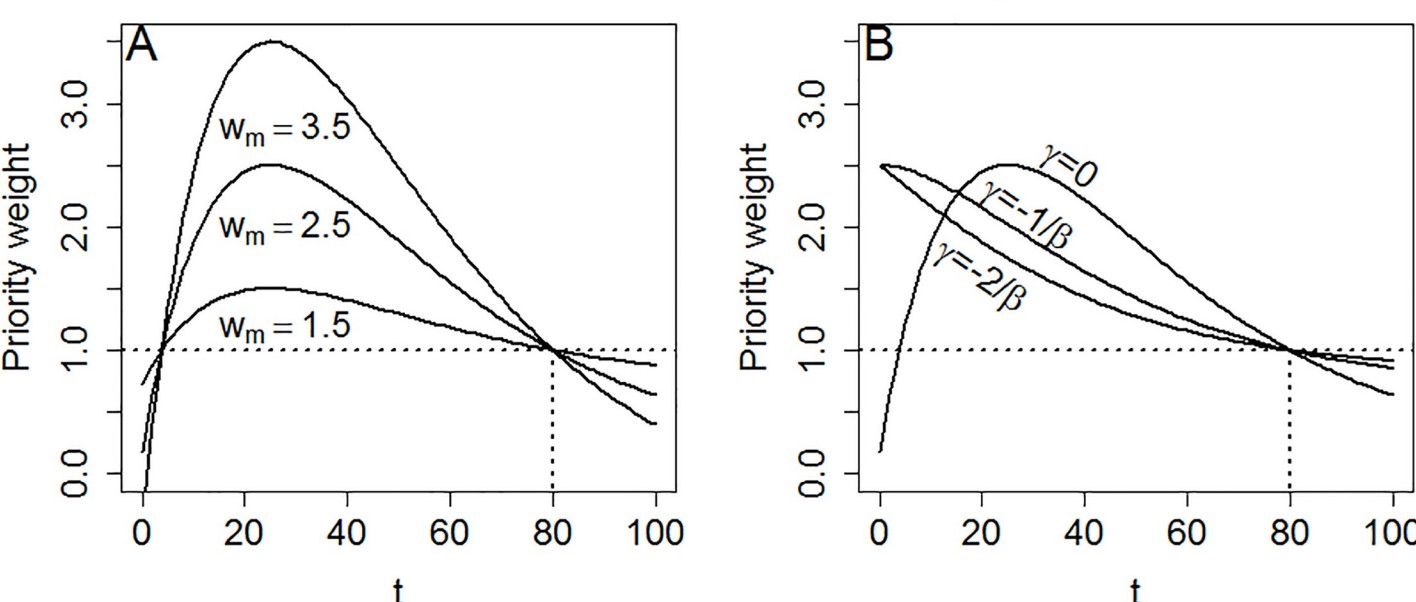

**Fig 3. Anchoring to 1 for t = 80, with $\beta$ = 0.04.** Panel A: Varying max t ($w_m$) when $\gamma$ = 0. Panel B: Varying $\gamma$ when $w_m$ = 2.5. For $\gamma$ = 0, there is no shift. For $\gamma = -1/\beta$ the curve is shifted left so that the maximum is at t = 0. For $\gamma = -2/\beta$ the curve is shifted so that the slope is the steepest at t = 0.

born with very severe conditions. In Figs 1 and 2 the default value of $\gamma$ was 0.04, so that the maximum was at t = 1/0.04 = 25 QALYs, which is probably too high in most scenarios. When $\gamma \leq -1/\beta$, a gain of, e.g., 1 QALY will always be weighted higher for the individual with the least lifetime health. From now on we shall assume that $\gamma \leq -1/\beta$.

Next we consider $\alpha$. The particular value of $\alpha$ is not necessarily easy to interpret, but because this coefficient allows for the anchoring of the PW at a certain weight for a certain value of t, it is useful when selecting a reference value of t for which PW = 1 (e.g., one may anchor PW at 1 when QALE = 70). Once a reference is chosen, the interpretation of the other coefficients is easier. A large β corresponds to a narrow peak of the PW, ensuring a steep decline of the curve when t is close to 0. Now, gains to individuals with little lifetime health are given much higher weights than gains to individuals with more lifetime health, whereas gains to individuals with intermediate and high levels of lifetime health will be weighted similarly. A small β yields a flat peak, so that the weight drops steadily as the initial health level improves (t increases). Although β also will affect the range of possible weights, this property is more closely connected to the C. The choice of C decides how much value will be assigned to a given t relative to the reference. E.g., one may think that the maximum weight should be no more than three times the weight of the reference. A very small C would ensure that all health gains were treated almost the same.

## Application of the priority weight function to experimental data

In order to illustrate the potential use of (5) in a CEA within a lifetime health framework, we fit the function to data from a convenience sample. The procedure involves two conceptually simple steps. First, data must be collected, and then the curve must be fit. However, both these steps may be performed a number of different ways. We shall focus on an approach that is suitable for a dataset presented in Ottersen et al. (2014).

Ottersen et al. used a computer based questionnaire to ask 96 students in Norway how they valued QALYs gained for people at different initial health levels, measured in "initial QALE" (QALE without intervention). The reference case was a group of healthy people with QALE = 70 who were subject to an intervention that would give them 10 extra QALYs (i.e., move from QALE = 70 to QALE = 80). Respondents were asked to assign QALYs to individuals with worse initial health (QALEs in the set {10, 25, 40, 55}) in such a way that they were indifferent between the different scenarios. Now, priority weights could be calculated for each individual respondent based on the ratio between the QALYs assigned to the unhealthy groups and the reference group. E.g., consider a respondent who was indifferent between a situation where the group with initial QALE = 25 gained 5 QALYs (i.e., moving from 25 to 30) and the reference group gained 10 QALYs (i.e., moving from 70 to 80). This would mean that gains in the group with a worse initial health, where initial QALE = 25, are regarded as more important than gains in the healthy group, where initial QALE = 70. Mathematically, this extra weight can be expressed as V(25) = 10/5 = 2 relative to the reference group (QALE = 70). We refer to Ottersen et al. for a more thorough discussion of data and study design.

One may organize the data as follows. Two lists, V and T, are constructed. V contains all the weights assigned by the respondents, and T contains the initial QALEs, so that the weight $V_i$ corresponds to the QALE $T_i$. For purposes of estimating the coefficients of (5), we treat the observations as independent. Applying a least squares approach, we get that the sum of squared errors (SSE) is

$$\text{SSE} = \sum\nolimits_i (V_i - \alpha + (T_i - \gamma) \cdot C \cdot e^{-\beta \cdot (T_i - \gamma)})^2. \tag{8}$$

This expression may be minimized using a numerical routine or by solving the set of equations

$$\frac{\partial}{\partial \alpha}\text{SSE} = 0, \frac{\partial}{\partial \gamma}\text{SSE} = 0, \frac{\partial}{\partial C}\text{SSE} = 0, \text{and} \frac{\partial}{\partial \beta}\text{SSE} = 0.$$

If one wishes to anchor the curve to a specific value (e.g., the data from Ottersen et al. has V (70) = 1), it is straightforward to use (6) in (8). Setting maximum weights would imply using (7) in (8). Also, any coefficient in (8) may be kept constant.

Minimizing SSE ensures that the curve is fit to the mean V for each T in {10, 25, 40, 55}. Using the R function optim() to minimize SSE, we get

$$\text{PW}_{\text{mean}} = -0.42 + (t + 22.2) \cdot 0.27 \cdot e^{-0.031 \cdot (t + 22.2)}. \tag{9}$$

However, minimizing instead the sum of absolute errors,

$$\text{SAE} = \sum_i |V_i - \alpha + (T_i - \gamma) \cdot C \cdot e^{-\beta \cdot (T_i - \gamma)}|, \tag{10}$$

the curve is fit to the median. Now PW becomes

$$\text{PW}_{\text{median}} = 0.79 + (t + 8.85) \cdot 0.17 \cdot e^{-0.053 \cdot (t + 8.85)}. \tag{11}$$

Note that when fitting the curves, $\alpha$ and $\gamma$ were set to ensure that PW(10) was the maximum (i.e., PW(10) = $w_m$) and PW(70) = 1. Fig 4 shows (9) and (11) plotted with the empirical data. Note that the $\text{PW}_{\text{mean}}$ is more affected by extreme values than are $\text{PW}_{\text{median}}$.

## Discussion

We have presented a new priority weight function (PW) with key characteristics that are important in consistent priority setting across patient groups and populations. As demonstrated, the PW was suitable for CEA in a lifetime health framework. The PW is flexible and easy to adjust to a broad range of equity considerations by altering a limited number of comprehensible parameters, each with its distinct role. We may set the range of priority weights by modifying C, we may shift the curve upwards or downwards by modifying $\alpha$, we may shift the curve left or right by modifying $\gamma$ and we may set the "pointiness" of the curve by modifying $\beta$. These characteristics make it easy, e.g., to anchor the curve or to set maximum weights. Considering (9) and (11) (Fig 4), we see that $\beta_{\text{median}} > \beta_{\text{mean}}$, indicating that $\text{PW}_{\text{median}}$ is pointier than $\text{PW}_{\text{mean}}$. By this we mean that for values of QALE around 10, $\text{PW}_{\text{median}}$ decreases at a faster rate relative to its maximum than does $\text{PW}_{\text{mean}}$. As QALE approaches 70, the relative rate of decrease is greater for $\text{PW}_{\text{mean}}$. Further, we have that $C_{\text{median}} < C_{\text{mean}}$, meaning that $\text{PW}_{\text{mean}}$ adds more weight to the worse off than does $\text{PW}_{\text{median}}$.

Drawing the PW by hand would of course be more flexible than using (5). However, we argue that setting some restrictions for the behaviour of the PW is useful. First, it offers consistency when comparing different curves. Second, the space of possible PWs is limited by (5), but the major restrictions caused by (5) seem reasonable (smooth and maximum one hump). Hence, a large number of nonsensical PWs are impossible (discontinuous and multiple humps).

We did not account for discounting of future health. It is not clear how this would affect rankings of interventions. A positive discount rate would put different weights on equally sized gains that were experienced at different times from the start of intervention (e.g., 0.5 QALYs gained immediately is valued higher than 0.1 QALYs gained each year in five years, or 0.5 QALYs five years from now). Hence, interventions with immediate effects would be valued

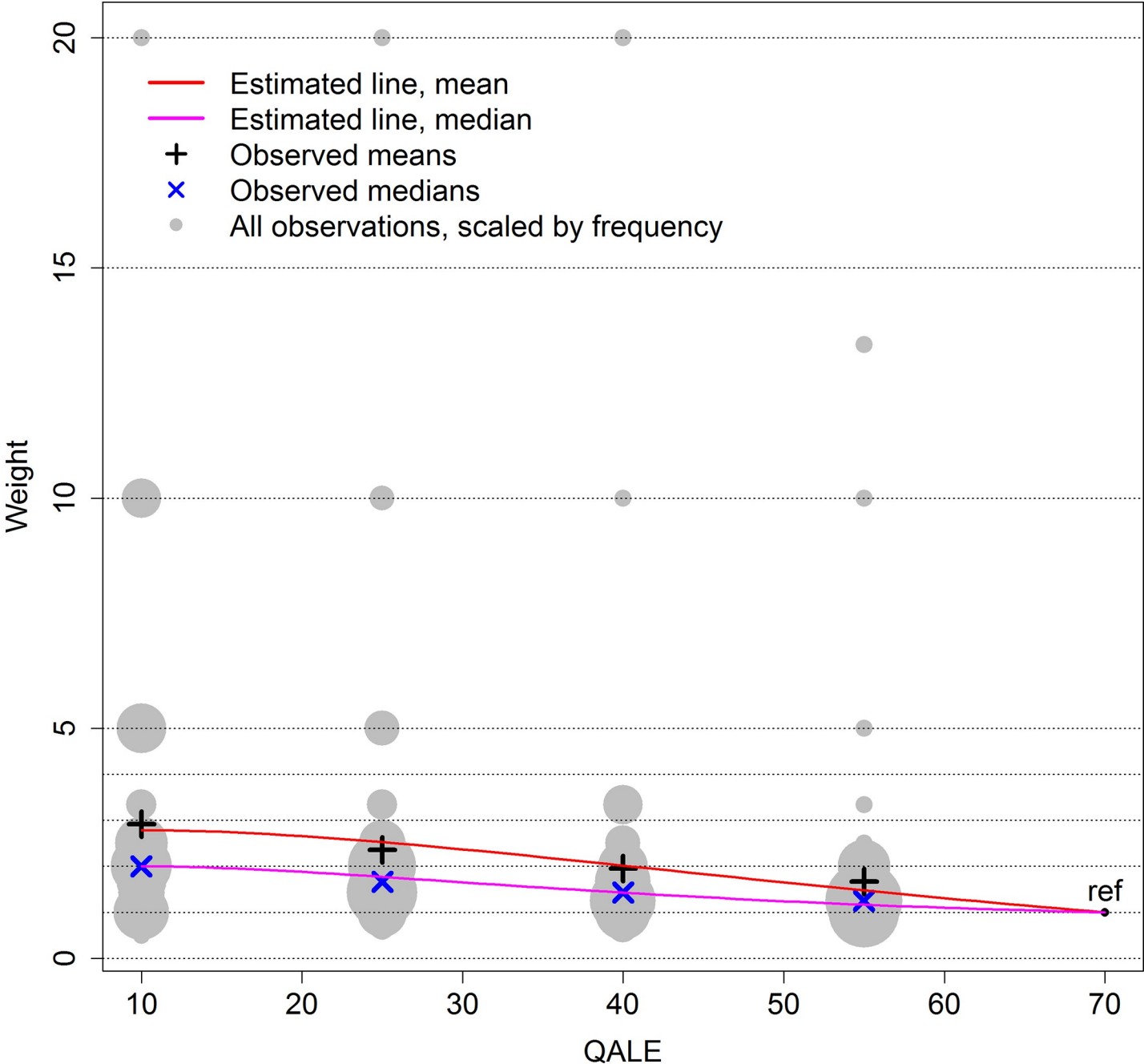

**Fig 4. Estimating PW from data.** The red line minimizes (8) to estimate a PW based on the means (black +). The coefficients are $\alpha = -0.42$, $\gamma = -22.2$, $C = 0.27$, and $\beta = 0.031$. The pink line minimizes (10) to estimate a PW based on the medians (blue ×). The coefficients are $\alpha = 0.79$, $\gamma = -8.85$, $C = 0.17$, and $\beta = 0.053$. In both scenarios $PW(10) = w_m$ (maximum weight at T = 10) and PW is anchored at $PW(70) = 1$.

over interventions where the benefit occurs in the future. Also, discounting health gains would give a relatively low priority to those with large gains, who are expected to have less lifetime health. This is counterintuitive in a lifetime health priority perspective.

A PW could be used theoretically to assess the impact of applying various ethical principles to rank health interventions, but ideally, the PW should be inferred from data. Although we do

give an example of how this can be done, future work includes developing methodology to do inference about parameter estimates based on empirical data. This would allow for the testing of differences in preferences between groups. E.g., conditioning on the other variables, one could test if men and women preferred different values of C (determining the maximum weight) or γ (determining the t for which the maximum weight was obtained). When an appropriate PW (or a range of PWs) has been selected, it could be applied to evaluate the equity impact of different interventions on reduction of mortality or morbidity. Also, work should be done to include non-health concerns like socioeconomic status and productivity in the model, and to explore the consequences of priority-weighted CEAs across settings.

In this paper we do not intend to determine which PW is the most appropriate, but rather to illustrate what can be achieved by a flexible general PW. However, we would like to stress the fact that a flat PW is also a PW. In other words, there is no such thing as "not using a PW" in CEA. As illustrated, our framework allows for the estimation of PWs based on empirical data.

## Acknowledgments

We would like to acknowledge Ole Frithjof Norheim and Trygve Ottersen for their useful comments during the creation of this manuscript.

## Author Contributions

**Conceptualization:** Øystein Ariansen Haaland, Frode Lindemark, Kjell Arne Johansson.

**Formal analysis:** Øystein Ariansen Haaland, Frode Lindemark, Kjell Arne Johansson.

**Investigation:** Øystein Ariansen Haaland, Frode Lindemark, Kjell Arne Johansson.

**Methodology:** Øystein Ariansen Haaland, Frode Lindemark, Kjell Arne Johansson.

**Project administration:** Øystein Ariansen Haaland, Kjell Arne Johansson.

**Software:** Øystein Ariansen Haaland, Frode Lindemark, Kjell Arne Johansson.

**Supervision:** Øystein Ariansen Haaland, Kjell Arne Johansson.

**Visualization:** Øystein Ariansen Haaland, Kjell Arne Johansson.

**Writing – original draft:** Øystein Ariansen Haaland, Frode Lindemark, Kjell Arne Johansson.

**Writing – review & editing:** Øystein Ariansen Haaland, Frode Lindemark, Kjell Arne Johansson.

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
