## [Decision Letter · Decision Letter 0]

19 Aug 2019

PONE-D-19-18169

A flexible formula for incorporating distributive concerns into cost-effectiveness analyses: priority weights

PLOS ONE

Dear Mr. Haaland,

Thank you for submitting your manuscript to PLOS ONE. After careful consideration, we feel that it has merit but does not fully meet PLOS ONE’s publication criteria as it currently stands. Therefore, we invite you to submit a revised version of the manuscript that addresses the points raised during the review process.

Please find below the reviewers' comments.

We would appreciate receiving your revised manuscript by Oct 03 2019 11:59PM. To enhance the reproducibility of your results, we recommend that if applicable you deposit your laboratory protocols in protocols.io, where a protocol can be assigned its own identifier (DOI) such that it can be cited independently in the future. For instructions see: http://journals.plos.org/plosone/s/submission-guidelines#loc-laboratory-protocols

We look forward to receiving your revised manuscript.

Kind regards,

Valerio Capraro

Academic Editor

PLOS ONE

Journal Requirements:

Additional Editor Comments:

I have now collected two reviews from two experts in the field. Both reviewers are positive towards the paper but suggest minor revisions. Therefore, I am glad to invite you to revise your paper following the reviewers' comments. Looking forward for the revision.

Reviewers' comments:

Reviewer's Responses to Questions

**Comments to the Author**

1. Is the manuscript technically sound, and do the data support the conclusions?

Reviewer #1: Yes

Reviewer #2: Yes

2. Has the statistical analysis been performed appropriately and rigorously? 

Reviewer #1: Yes

Reviewer #2: I Don't Know

3. Have the authors made all data underlying the findings in their manuscript fully available?

Reviewer #1: Yes

Reviewer #2: Yes

4. Is the manuscript presented in an intelligible fashion and written in standard English?

Reviewer #1: Yes

Reviewer #2: Yes

5. Review Comments to the Author

Reviewer #1: The authors of this paper have proposed a flexible priority weight function that is designed to incorporate concerns about the distribution of lifetime health across individuals into cost-effectiveness analysis. I appreciate the contribution the authors have made. I have a few suggestions and questions below.

Additional clarity in the abstract may help readers better understand the contribution. First, it’s not clear what is meant by ‘similar health characteristics’. Is this measured using preference-weighted measures of health, like a utility weight corresponding to a given health state, at a moment in time, or does it account for accumulated health? Second, in the following sentence, the authors state that the function if flexible regarding shape and ‘outcome measures’. Do those outcome measures reflect health outcomes measured over the remaining lifetime for patients?

Also, I would prefer that the authors do not include the PWmean and PWmedian functions with weights in the abstract as those weights should not be construed as generalizable. They are based on a small convenience sample of students.

Unfortunately, there are no page numbers on the printed version of the paper.

On the first page after the abstract, line 34, the authors state that “… those that are worse off over slightly less cost-effective interventions”. I would have expected this to be ‘more cost-effective interventions’ to demonstrate that people generally prioritize less efficient investments in those that are worse off. Please confirm.

In the paragraph after [Figure 1 here], it is stated ‘in Figure 2B, the maximum can be moved beyond t=0 making (7) strictly decreasing’. It appears that when ϒ is -40, the maximum PW is at t=0 with decreasing PW. But, what is meant by ‘beyond t=0’? less than t? I note that later in the paper, it is stated that letting ϒ= -1/β ensures that the maximum is at t=0, so that seems to be a better way to ensure that the maximum PW is at 0.

In the paragraph after [Figure 2 here] ‘if we want to ensure that people with a certain value of t are given a certain weight (anchoring the curve), we may simply change α’. It appears to me that Figures 3A and 3B represent cases when the PW is anchored at t=80, indicating that one would want to vary Wm or ϒ. It appears that varying α would change the PWs across different values of t. Please try to explain this more clearly.

In the section on ‘Application of the priority weight function to experimental data’, the questionnaire described for the Ottersen et al. study does not seem to represent a discete-choice experiment wherein respondents were asked to choose which QALE and QALY gain scenarios they prefer. Instead, respondents were asked to assign QALYs with varying levels of initial health such that they would be indifferent. This weighting exercise does not represent a DCE. Please explain.

In the Discussion, lines 14-16, the authors point out that βmedian is greater than βmean indicating that PWmedian decreases at a faster rate than PWmean. In my view, according to Figure 4, it appears that PWmean decreases at a faster rate than PWmedian as PWmean starts at a higher value and both lines converge. Please explain. It is difficult to discern that PWmedian is ‘pointier’.

Reviewer #2: The authors address an important question which is well motivated by the background. I have comments about the communication and presentation that are not critiques of the overall approach.

While in general I appreciate the specificity, presenting the equation in the abstract without defining the variables in unclear, it is impossible for readers to interpret.

There should be clearer summary or conclusions in the abstract that are more detailed. The final sentence in the abstract is too general to be useful.

At the end of the second to last paragraph of the introduction, the authors discuss the relevance of age to QALE calculation. I found this to be seemingly important but the relevance was unclear. Perhaps the authors can rewrite this section.

The authors refer to another paper by Rodriquez and Pinto for discussion of u(t), but it would be useful to have more explanation here since it seems to be important for estimates.

It would be useful to answer the question of why we need a mathematical function at all, why not a linear slope?

Do the authors suggest that investigators pick values and then look at the curve, is the overall idea to find a curve that makes sense, or values that make sense, and should these be done a priori to looking at the data?

In figure 2, it seems unusual that the function should be lower at 0 and close to 0, is this intentional and/or desirable? It would be useful to have text explaining this.

6. PLOS authors have the option to publish the peer review history of their article (what does this mean?). If published, this will include your full peer review and any attached files.

Reviewer #1: No

Reviewer #2: No

---

## [Author Response · Author response to Decision Letter 0]

12 Sep 2019

Response to reviewers. 

All page numbers and line numbers refer to the manuscript with tracked changes. 

Initial comment from authors: The equation numbers were wrong, and we have changed these throughout the manuscript. We have also done some other minor changes to improve readability. These can be found in the manuscript with tracked changes. 

Reviewer #1: 

Additional clarity in the abstract may help readers better understand the contribution. 

Comment: First, it’s not clear what is meant by ‘similar health characteristics’. Is this measured using preference-weighted measures of health, like a utility weight corresponding to a given health state, at a moment in time, or does it account for accumulated health? 

Reply: We have added the following to the second paragraph of the abstract (lines 13-15 on page 2): 

“For example, those who achieve 50 QALE should be treated similarly to those who achieve 49.9 QALE.” 

We have also defined QALE at the end of the first paragraph (lines 7-8 on page 2). 

Comment: Second, in the following sentence, the authors state that the function is flexible regarding shape and ‘outcome measures’. Do those outcome measures reflect health outcomes measured over the remaining lifetime for patients?

Reply: We have added the following to the Abstract (line 15 on page 2): 

“(i.e., caters to other measures than QALE)” 

and the last paragraph of the Introduction (lines 18-19 on page 4):

“(e.g., cater to other measures than QALE)”

Comment: Also, I would prefer that the authors do not include the PWmean and PWmedian functions with weights in the abstract as those weights should not be construed as generalizable. They are based on a small convenience sample of students.

Reply: We agree, and have deleted lines 20-23 on page 2. 

Comment: Unfortunately, there are no page numbers on the printed version of the paper.

Reply: We are so sorry about that! Page numbers have been added!

Comment: On the first page after the abstract, line 34, the authors state that “… those that are worse off over slightly less cost-effective interventions”. I would have expected this to be ‘more cost-effective interventions’ to demonstrate that people generally prioritize less efficient investments in those that are worse off. Please confirm.

Reply: Of course, the reviewer is right. We have changed “less” to “more”, as suggested (line 34 on page 3). 

Comment: In the paragraph after [Figure 1 here], it is stated ‘in Figure 2B, the maximum can be moved beyond t=0 making (7) strictly decreasing’. It appears that when ϒ is -40, the maximum PW is at t=0 with decreasing PW. But, what is meant by ‘beyond t=0’? less than t? I note that later in the paper, it is stated that letting ϒ= -1/β ensures that the maximum is at t=0, so that seems to be a better way to ensure that the maximum PW is at 0.

Reply: The relevant text now says (lines 11 and 12 on page 6): 

“As is shown in Figure 2B, it is possible to ensure that the maximum is at t=0, making (5) strictly decreasing.” 

Comment: In the paragraph after [Figure 2 here] ‘if we want to ensure that people with a certain value of t are given a certain weight (anchoring the curve), we may simply change α’. It appears to me that Figures 3A and 3B represent cases when the PW is anchored at t=80, indicating that one would want to vary Wm or ϒ. It appears that varying α would change the PWs across different values of t. Please try to explain this more clearly.

Reply: We thank the reviewer for pointing out that this was not clear, and have added a paragraph on lines 15-18 on page 7: 

“Note that although varying α without changing the other coefficients just shifts PW_1 up or down without changing its shape (Figure 2A), fixing the other coefficients in this manner is not something one would typically do. In order to keep predetermined properties of PW_1, such as anchoring, a change in α would prompt a change in one or more of the other coefficients.”

Comment: In the section on ‘Application of the priority weight function to experimental data’, the questionnaire described for the Ottersen et al. study does not seem to represent a discete-choice experiment wherein respondents were asked to choose which QALE and QALY gain scenarios they prefer. Instead, respondents were asked to assign QALYs with varying levels of initial health such that they would be indifferent. This weighting exercise does not represent a DCE. Please explain.

Reply: We thank the reviewer for this comment. We have changed the text in Application of the priority weight function to experimental data (lines 6-7 on page 8) to the following: 

“In order to illustrate the potential use of (5) in a CEA within a lifetime health framework, we fit the function to data from a convenience sample.” 

We also changed the text in the Discussion (lines 17-21 on page 10) to the following: 

“A PW could be used theoretically to assess the impact of applying various ethical principles to rank health interventions, but ideally, the PW should be inferred from data. Although we do give an example of how this can be done, future work includes developing methodology to do inference about parameter estimates based on empirical data.”

Comment: In the Discussion, lines 14-16, the authors point out that βmedian is greater than βmean indicating that PWmedian decreases at a faster rate than PWmean. In my view, according to Figure 4, it appears that PWmean decreases at a faster rate than PWmedian as PWmean starts at a higher value and both lines converge. Please explain. It is difficult to discern that PWmedian is ‘pointier’.

Reply: We have changed the relevant text (lines 21 on page 9 - line 1 on page 10) to: 

“Considering (9) and (11) (Figure 4), we see that β_median>β_mean, indicating that PWmedian is pointier than PWmean. By this we mean that for values of QALE around 10, PWmedian decreases at a faster rate relative to its maximum than does PWmean. As QALE approaches 70, the relative rate of decrease is greater for PWmean.” 

Further, we have changed lines 34-35 on page 7 to:

“A large β corresponds to a narrow peak of the PW, ensuring a steep decline of the curve when t is close to 0.” 

Reviewer #2: 

Comment: While in general I appreciate the specificity, presenting the equation in the abstract without defining the variables in unclear, it is impossible for readers to interpret.

Reply: See reply to Reviewer 1

Comment: There should be clearer summary or conclusions in the abstract that are more detailed. The final sentence in the abstract is too general to be useful.

Reply: We have revised the last sentence of the abstract (lines 24-27 on page 2) as follows: 

“Equity concerns are important when conducting CEAs, which means that suitable PWs should be developed. We do not intend to determine which PW is the most appropriate, but to illustrate how a flexible general PW can be estimated based on empirical data.” 

Comment: At the end of the second to last paragraph of the introduction, the authors discuss the relevance of age to QALE calculation. I found this to be seemingly important but the relevance was unclear. Perhaps the authors can rewrite this section.

Reply: We have tried to clarify the text (lines 9-13 on page 4): 

“Note that although QALE often increases with age, a high age does not necessarily correspond to a high QALE. E.g., the QALE of an old person who has experienced decades of chronic illness may be far less than that of a healthy child. In other words, current age is not a good measure of QALE, and weights based on age are therefore different than weights based on lifetime health.”

Comment: The authors refer to another paper by Rodriquez and Pinto for discussion of u(t), but it would be useful to have more explanation here since it seems to be important for estimates.

Reply: We thank the reviewer for this comment. The brief review of u(t) and the social welfare functions is meant to motivate the development of PW, but we do not use them directly. Currently, we write: 

“However, u(t) does not account for the distribution of lifetime health. We approach this problem by considering priority weight functions (PWs) instead.”

If this is not clear enough, we would be happy to rewrite the text. However, we prefer not to include more information about u(t), because u(t) cannot be used in a lifetime health perspective, which is the focus of this paper. At one point in the preparation of the manuscript, we did discuss u(t) in more detail, but internal reviewers found it confusing. 

Comment: It would be useful to answer the question of why we need a mathematical function at all, why not a linear slope?

Reply: This is an important point, and we have added the following text where we discuss flexibility in Methods (lines 21-23 on page 5): 

“A lack of flexibility would also be a problem if one would use a linear PW, such as 

PW_linear (t_i)=〖a+b⋅t〗_i. (4)”

We also add the following in the next paragraph (lines 31-36 on page 5): 

“While these concerns could be addressed using (4), a linear PW falls short in other aspects. E.g., a hump is useful if one wants to give lower weights to health gains benefitting those with a very low QALE (typically children born with severe conditions). If one wishes to do the opposite, and prioritize health gains to those who are expected to achieve very little lifetime health, PW may peak at t=0, and then drop at a faster rate when t is close to 0 than when t is higher.” 

Finally, we have changed the text on lines 21-25 on page 7: 

“First, we consider γ. If γ>-1/β, it is possible that PW(t_1)<PW(t_2) for t_1<t_2. This makes sense if one thinks that small gains to individuals with almost no lifetime health should be worth less than a similar gain to an individual with more lifetime health. As mentioned, this could pertain to treatment of children born with very severe conditions.”

Comment: Do the authors suggest that investigators pick values and then look at the curve, is the overall idea to find a curve that makes sense, or values that make sense, and should these be done a priori to looking at the data?

Reply: We thank the reviewer for this comment. We have done the following changes in the Discussion (lines 1-4 on page 10): 

“A PW could be used theoretically to assess the impact of applying various ethical principles to rank health interventions, but ideally, the PW should be inferred from data. Although we do give an example of how this can be done, future work includes developing methodology to do inference about parameter estimates based on empirical data.” 

Comment: In figure 2, it seems unusual that the function should be lower at 0 and close to 0, is this intentional and/or desirable? It would be useful to have text explaining this.

Reply: We thank the reviewer for this comment, and have added the following to the figure text of Figure 2A (line 8 on page 13): 

“Note that one would typically chose a PW where the maximum weight was at zero.”

---

## [Decision Letter · Decision Letter 1]

1 Oct 2019

A flexible formula for incorporating distributive concerns into cost-effectiveness analyses: priority weights

PONE-D-19-18169R1

Dear Dr. Haaland,

We are pleased to inform you that your manuscript has been judged scientifically suitable for publication and will be formally accepted for publication once it complies with all outstanding technical requirements.

With kind regards,

Valerio Capraro

Academic Editor

PLOS ONE

Additional Editor Comments (optional):

Reviewers' comments:

Reviewer's Responses to Questions

**Comments to the Author**

1. If the authors have adequately addressed your comments raised in a previous round of review and you feel that this manuscript is now acceptable for publication, you may indicate that here to bypass the “Comments to the Author” section, enter your conflict of interest statement in the “Confidential to Editor” section, and submit your "Accept" recommendation.

Reviewer #1: All comments have been addressed

Reviewer #2: All comments have been addressed

2. Is the manuscript technically sound, and do the data support the conclusions?

Reviewer #1: Yes

Reviewer #2: Yes

3. Has the statistical analysis been performed appropriately and rigorously? 

Reviewer #1: Yes

Reviewer #2: I Don't Know

4. Have the authors made all data underlying the findings in their manuscript fully available?

Reviewer #1: Yes

Reviewer #2: Yes

5. Is the manuscript presented in an intelligible fashion and written in standard English?

Reviewer #1: Yes

Reviewer #2: Yes

6. Review Comments to the Author

Reviewer #1: (No Response)

Reviewer #2: Thank you for addressing all of my previous comments.

7. PLOS authors have the option to publish the peer review history of their article (what does this mean?). If published, this will include your full peer review and any attached files.

Reviewer #1: No

Reviewer #2: No

---

## [Editor Report · Acceptance letter]

2 Oct 2019

PONE-D-19-18169R1 

A flexible formula for incorporating distributive concerns into cost-effectiveness analyses: priority weights 

Dear Dr. Haaland:

I am pleased to inform you that your manuscript has been deemed suitable for publication in PLOS ONE. Congratulations! Your manuscript is now with our production department. 

With kind regards,

on behalf of

Dr. Valerio Capraro 

Academic Editor

PLOS ONE